# Steviol Glycosides Supplementation Affects Lipid Metabolism in High-Fat Fed STZ-Induced Diabetic Rats

**DOI:** 10.3390/nu13010112

**Published:** 2020-12-30

**Authors:** Jakub Michał Kurek, Ewelina Król, Zbigniew Krejpcio

**Affiliations:** Department of Human Nutrition and Dietetics, Poznań University of Life Sciences, Wojska Polskiego 31, 60-624 Poznań, Poland; jakub.kurek@up.poznan.pl (J.M.K.); ewelina.krol@up.poznan.pl (E.K.)

**Keywords:** steviol glycosides, supplementation, high-fat diet, diabetes, rats

## Abstract

A number of health-promoting properties of *Stevia rebaudiana* Bertoni and its glycosides, including the antihyperglycemic activity, have been found. The mechanisms of the antidiabetic action of stevia have not been fully understood. The aim of this study was to evaluate the effects of supplementary steviol glycosides on high-fat fed streptozotocin-induced diabetic rats with particular attention to lipid metabolism. The experiment was conducted on 70 male Wistar rats, of which 60 were fed a high-fat diet for 8 weeks followed by intraperitoneal injection of streptozotocin, to induce type 2 diabetes. Afterwards, rats were divided into six groups and fed a high-fat diet supplemented with pure stevioside or rebaudioside A, at two levels (500 or 2500 mg/kg body weight (b.w.)) for 5 weeks. Three additional groups: diabetic untreated, diabetic treated with metformin, and healthy, served as respective controls. Blood and dissected internal organs were collected for hematological, biochemical, and histopathological tests. It was found that dietary supplementation with steviol glycosides did not affect blood glucose, insulin, and insulin resistance indices, antioxidant biomarkers, but normalized hyperlipidemia and affected the appetite, as well as attenuated blood liver and kidney function indices, and reduced tissular damage in diabetic rats. Steviol glycosides normalize lipid metabolism and attenuate internal organs damage in diabetes.

## 1. Introduction

Diabetes mellitus (DM) is one of the most prevalent lifestyle diseases, with an increasing impact on human health. The International Diabetes Federation reports that nearly 10% of the world population suffers from DM and its complications, resulting in ca. 4.2 million fatalities in 2019 [1]. The development of diabetes is associated with genes and family history, however, according to the National Institute of Diabetes and Digestive and Kidney Diseases, the main risk factors for type 2 diabetes (T2DM) are excessive body weight and a lack of physical activity [2,3]. The immediate cause of the disease is insulin resistance, a condition in which the body’s cells are unable to use insulin efficiently. This results in excessive insulin secretion until the pancreas is no longer able to keep up with the body’s needs, which is followed by chronic hyperglycemia. The effects of untreated T2DM are life-threatening macrovascular and microvascular complications, such as: atherosclerosis, coronary heart disease, stroke, retinopathy, nephropathy, and neuropathy, respectively. Currently available treatments that help control glycemia and slow down diabetic complications include insulin and various oral antidiabetic agents, such as sulfonylureas, biguanides, and β-glucosidase inhibitors. On the other hand, they cause various side effects [4,5]. Diets and some food supplements can also contribute to control hyperglycemia and reduce diabetic complications [6].

Raman et al. (2012) report that up to 30% of patients with diabetes mellitus use complementary and alternative medicine including medicinal plants and its products (food and supplements) [7]. Therefore, there is a search for complementary and alternative medicine that involves the use of various plants, herbs, and other dietary supplements.

One such plant is sweet herb *Stevia rebaudiana* Bertoni, which belongs to the *Compositae* family and is native to parts of South America, especially Brazil and Paraguay.

*Stevia rebaudiana*, a natural, non-caloric sweetener, has generated significant interest in the scientific community due to its effects on glucose homeostasis, blood pressure, and inflammation—all known consequences of obesity. Purified compounds are steviol glycosides (SGs), stevioside (St), rebaudioside A (RA), and several other, are Generally Recognized as Safe (GRAS) by the Food and Drug Administration (FDA) [8]. These compounds have been approved by European Food Safety Authority (EFSA) as intense sweeteners (E960) in the food industry. 

A number of scientific reports showed that stevia preparations or its isolated compounds can exert several health-promoting effects, for instance: antimicrobial/antifungal [9], antioxidant [10], antihypertensive [11], and anti-inflammatory [12], anti-caries [13], and antidiabetic activity [14]. 

Concerning the antidiabetic activity of stevia and its derivatives, scientific reports have shown inconsistent results, due to using different fractions of stevia or its extracts or compounds, doses, duration of treatment, as well as the experimental models (in vitro, in vivo, animal study, human trials). For example, Saravanan and Ramachandran (2013) examined the effects of rebaudioside A on blood glucose and insulin levels, lipid peroxidation, antioxidative activity, and lipid profile during a 45-day experiment on healthy and diabetic Wistar rats [15]. It was found that the treatment with rebaudioside A improved blood glucose and insulin levels in diabetic rats. Moreover, lipid profile, lipid peroxidation biomarkers in the blood plasma and the levels of enzymatic and non-enzymatic antioxidants were comparable to the parameters of healthy rats. This experiment confirmed the beneficial effects of treatment with rebaudioside A on glucose management and lipid metabolism in diabetes. The antidiabetic potential of stevia and its compounds was recently reviewed [16,17].

The aim of the study was to investigate the effects of supplementary steviol glycosides (stevioside and rebaudioside A) on type 2 diabetic rats, with particular attention to lipid metabolism.

## 2. Materials and Methods 

### 2.1. Test Supplements

Metformin (Met), in the form of metformin hydrochloride, was obtained from a commercially available antidiabetic drug, Metformax (Teva Operations Poland Ltd., Kraków, Poland). 

Stevioside (St) and rebaudioside A (RA) (≥98%, HPLC) were purchased from Anhui Minmetals Development Co., Hefei, China. Streptozotocin (STZ) (≥98%, HPLC) was purchased from Sigma-Aldrich Ltd., Poznań, Poland.

### 2.2. Animals and Diets

This 13-week experiment was conducted on 70 males of Wistar Rat (*Rattus norvegicus*, outbred strain) at the age of 6 weeks and initial body weight of 222.5 ± 19.5 g. The numbers of animals per experimental group was established based on a standard approach calculation (an expected blood glucose level) as follows: for a two-sided t-test with alpha = 0.05 and the standard deviation of 0.25 (25%) for the power of 0.80, the sample size is 9. However, we used 10 animals per experimental group, taking into account a possible loss of diseased animals (10% attrition). The animals were obtained from Charles River Laboratories, Inc., Sulzfeld, Germany and supplied by Animalab Ltd., Poznań, Poland. The experiment was carried out in Association for Assessment and Accreditation of Laboratory Animal Care International (AAALAC) approved Animal Care Facility in Poznań University of Life Sciences, with controlled conditions: the ambient air temperature at 18 ± 1 °C and humidity at 57 ± 3%., photoperiod light/dark 12/12 h.

The diets were prepared by Urszula Borgiasz Zoolab, Sędziszów, Poland. The diets were given in the form of pellets with a diameter of 10–12 mm. The animals had unlimited access to food and clean drinking water (ad libitum). The control group (C) received the AIN-93M standard maintenance diet throughout the entire experiment. The diabetic group (Db) received a high-fat (HF) diet (40% energy from fat). The diabetic group Db + Met received a HF diet enriched with metformin (0.15%, equals to 150 mg/kg b.w./day). Other diabetic groups, i.e., Db + S1, Db + S2, Db + R1, Db + R2 received HF diets enriched with SGs (St or RA) in two different doses (0.5% or 2.5%, equals to 500 or 2500 mg/kg b.w./day). The allocation of rats to individual groups is presented in Figure 1. The rats were weighed once a week, while their feed intake was monitored daily.

### 2.3. Experimental Protocol

The experiment began with a one-week adaptation period during which the rats acclimatized to the conditions in the animal facility. Afterward, animals were divided into two groups: control (C) (healthy rats, *n* = 10) fed with the AIN-93M diet and experimental (diabetic rats, *n* = 60), fed with the HF diet in order to induce insulin resistance. After 8 weeks, intraperitoneal STZ injection (35 mg/kg b.w., dissolved in 0.1 M citrate buffer, pH 4.4) was performed on the experimental group to induce hyperglycemia, while the control (C) group received a placebo injection (0.1 M citrate buffer, pH 4.4, ≤1 mL/kg b.w.). After confirmation of hyperglycemia with iXell glucose meter (Genexo, Warsaw, Poland), the rats from the experimental group (*n* = 60) were divided into 6 groups (10 rats each), and then a five-week supplementation period was introduced, according to the scheme presented in Figure 1. In the course of the experiment, six rats died due to severe diabetic complications.

At the end of the experiment, all animals were sacrificed by decapitation (after 5-h fasting), blood was collected and internal organs were dissected, washed in 0.9% saline, weighed, then snapshot frozen in liquid nitrogen, then transferred to the deep freezer (−80 °C) and stored until further analyses. The whole blood and separated serum were immediately transported to ALAB laboratories (Poznań, Poland) for adequate analyses (blood hematological and biochemical indices). For histopathological examination, properly truncated selected organs (liver, kidneys, and pancreas) were placed in 10% buffered formalin and transported to ALAB laboratories (Poznań, Poland). The experimental protocol was approved by the Local Ethical Commission in Poznań (No. 31/2019).

### 2.4. Blood Hematological Analysis

The blood hematological analyses included the following indices: red blood cell (RBC (T/l)), blood hemoglobin (HGB (g/dl)), hematocrit (HCT (%)), mean corpuscular volume (MCV (fl)), mean corpuscular hemoglobin (MCH (pg)), mean corpuscular hemoglobin concentration (MCHC (g/dL)), platelets (PLT (G/l)), red blood cell distribution width (RDW-CV (%)), white blood cells (WBC (G/l)), neutrophils (NEUT% (%)/NEUT (G/l)), lymphocytes (LYM% (%)/LYM (G/l)), monocytes (MONO% (%)/MONO (G/l)), eosinophils (EOS% (%)/EOS (G/l)), and basophils (BAS% (%)) that were determined using the BC-5000 Vet Automatic Hematology Analyzer (Mindray Medical Poland Ltd., Warsaw, Poland).

### 2.5. Serum Biochemical Analysis

The blood serum biochemical analyses were conducted using adequate methods, and included the following indices: total cholesterol (TC (mg/dL)) was determined using the Konelab 20i Analyzer (Thermo Fisher Scientific Corporation, Vantaa, Finland); alanine transaminase (ALT (U/l)) was determined by the IFCC method with pyridoxal phosphate activation [18,19]; aspartate transaminase (AST (U/l)) by the IFCC method without pyridoxal phosphate activation [20,21]; urea concentration (UREA (mg/dL)) by the kinetic method with urease and glutamate dehydrogenase [22,23,24,25]; creatinine concentration (KREA (mg/dL)) by the kinetic colorimetric assay based on the Jaffé method [26,27,28]; glucose concentration (GLU (mg/dL)) was determined by the enzymatic reference method with hexokinase [29,30]; total protein (TP level (g/dl)) by the colorimetric assay using TP2 [31]; triacylglycerols level (TG (mg/dL)) by the enzymatic colorimetric method [32]; uric acid concentration (URIC (mg/dL)) by the enzymatic colorimetric method [33]. The lipid profile markers: total cholesterol concentration (TC (mg/dL)) was determined by the standard enzymatic method (Thermo Fisher Scientific, Waltham, USA) [34]; LDL cholesterol level (LDL (mg/dL)) by the homogeneous enzymatic colorimetric method [35,36]; HDL cholesterol concentration (HDL [mg/dL]) by the homogeneous enzymatic colorimetric method [37,38]. The levels of oxidized low-density lipoprotein cholesterol (OxLDL [µg/mL]), nitric oxide (NO (µmol/L)), insulin (INS (nIU/l)), ghrelin (ghrelin (ng/mL)), leptin (leptin (pg/mL)), adiponectin (adiponectin (ng/mL)), glutathione peroxidase activity (GPX (nmol/min/mL)), were determined using specific commercially available ELISA kits (Shanghai Sunred Biological Technology Co., Ltd., Shanghai, China; CUSABIO TECHNOLOGY LLC, Houston, TX, USA; EMD Millipore Corporation, Billerica, MA, USA; EIAab Science Company, Wuhan, China; Cayman Chemical Company, Ann Arbor, MI, USA). Enzyme-linked immunosorbent assays were performed using a Thermo Scientific Multiskan GO UV/VIS spectrophotometer (Thermo Fisher Scientific Corporation, Vantaa, Finland) or an ASYS UVM 340 plate reader (Asys Hitech, Cambridge, UK). Other blood biochemical indices were determined using a Cobas Integra400+ multifunctional biochemical analyzer (Roche Holding AG, Basel, Switzerland, 2019).

### 2.6. The Formulas for Calculation of FER, HOMA-IR, HOMA-β and QUICKI Indices

Feeding efficiency ratio (FER) was calculated using the following formula: (1)$$FER=\frac{total\;body\;weight\;gain\;\left[g\right]\;\times \;100}{overall\;feed\;intake\;\left[g\right]}$$

Insulin resistance and β-cell function were evaluated by the Homeostasis Model Assessment Method. HOMA-IR was calculated using the following formula: (2)$$\mathrm{HOMA}-\mathrm{IR}=\frac{\mathrm{fasting}\;\mathrm{glucose}\;\left[\mathrm{mmol}/L\right]\times \mathrm{fasting}\;\mathrm{insulinemia}\;\left[\mu \mathrm{IU}/\mathrm{mL}\right]}{22.5}$$

HOMA-β was calculated using the following formula: (3)$$\mathrm{HOMA}-\beta =\frac{20\;\times \;\mathrm{fasting}\;\mathrm{insulin}\;\left[\mu \mathrm{IU}/\mathrm{mL}\right]}{\mathrm{fasting}\;\mathrm{glucose}\;\left[\mathrm{mmol}/\mathrm{mL}\right]-3.5}$$

The quantitative insulin sensitivity check index (QUICKI) was also evaluated and calculated using the following formula: (4)$$\mathrm{QUICKI}=\frac{1}{\log\;\left(\mathrm{fasting}\;\mathrm{glucose}\;\left[\mathrm{mg}/\mathrm{dL}\right]\right)+\log\;\left(\mathrm{fasting}\;\mathrm{insulin}\;\left[\mathrm{microU}/\mathrm{mL}\right]\right)}$$

### 2.7. Histopathological Analyses

Dissected fragments of organs: pancreas, liver and kidneys were put in 10% buffered formalin, then subjected to histological preparation. The truncated organs were placed in histological cassettes. The material was tested in a tissue processor with a series of alcohols and xylene to paraffin—in accordance with standard protocols of histological paraffin techniques. Tissues embedded in paraffin blocks were cut using a histological microtome. Tissues were stained with standard hematoxylin-eosin topographic staining. The histopathological evaluation included 3–4 sections from each organ. As part of the histopathological evaluation, a scalar evaluation was made of the severity of parameters typical for STZ-induced diabetes. As part of the assessments, a description of additional identified changes not included in the scoring assessment was made.

Histopathological evaluation was performed by 3 independent pathologists, using relevant criteria based on guidelines of International Harmonization of Nomenclature and Diagnostic Criteria (INHAND) elaborated by ESTP, STP, BSTP, JSTP and OECD general and histopathological recommendations and protocols of the National Toxicology Program for pathological assessments [39,40]. 

### 2.8. Statistical Analyses

The results were statistically evaluated using Microsoft Excel 2019 (Microsoft Corporation, Redmond, USA) and Statistica 13.3 (TIBCO Software Inc., Palo Alto, CA, USA). The analysis of variance (ANOVA) and Fisher’s Least Significant Difference (LSD) post hoc test (*p <* 0.05) were used to evaluate the level of statistical significance of differences in individual parameters between the groups. The Shapiro–Wilk test was used as a normality test of data sets. All data are expressed as mean ± standard deviation (M ± SD). The data are presented in tables and figures. The statistically significant differences in values between experimental groups were marked with letters: a, b, c, etc. Those having different letters are significantly different from each other’s (*p <* 0.05, a < b, Fisher’s LSD test), while those sharing common letters are not different; i.e., “ab” it is not different from either “a” nor “b”, but it is different from “c”.

## 3. Results

### 3.1. Effects of Supplementary SGs on Overall Growth Indices and Relative Organ Masses

Before induction of hyperglycemia (by STZ injection), rats fed the HF diet had significantly lower feed intake (24.61 ± 1.00 vs. 29.22 ± 1.40 g/day/rat, *p <* 0.05), but higher body mass gain (281.0 ± 28.57 vs. 246.5 ± 39.28 g/rat, *p* < 0.05) and higher FER (11.89 ± 1.08 vs. 8.77 ± 1.21 g body mass/g diet *100, *p* < 0.05), as compared with the healthy control fed standard AIN-93M.

Table 1 shows the results of the supplementary stage of the experiment. After STZ injection, the diabetic rats had significantly higher average feed intake (29.94 ± 3.94 vs. 27.20 ± 1.61 g/day/rat, *p* < 0.05), but markedly lower body mass gain (−3.00 ± 44.33 vs. 44.70 ± 12.68, g/rat, *p* < 0.05). Final body mass was significantly lower (473.4 ± 46.86 vs. 543.4 ± 51.50 g/rat, *p* < 0.05), FER decreased (0.02 ± 1.67 vs. 1.71 ± 0.46 body mass/g diet *100, *p* < 0.05), as compared with the control group. The relative internal organ masses were increased in a variable degree, as compared with the healthy control rats, but in most cases, it was the result of a decreased body mass gain or loss, not organ enlargement itself, except the kidneys and the lung masses, which were significantly higher (0.95 ± 0.17 vs. 0.63 ± 0.07 %, *p* < 0.05 and 0.41 ± 0.07 vs. 0.32 ± 0.04 %, *p* < 0.05, respectively), as compared with the healthy control rats.

Table 1 shows that supplementary stevioside and rebaudioside A at high dose (2500 mg/kg b.w.) markedly lowered feed intake (by 10%), almost to the level of the healthy control rats.

### 3.2. Effects of Supplementary SGs on Lipid Metabolism

The effect of experimental factors on the lipid profile indices: serum TG, TC, LDL and HDL are presented in Figure 2. As expected, the diabetic untreated rats (Db) had significantly elevated serum lipid biomarkers, TG (635.36 ± 331.6 vs. 246.30 ± 97.3 mg/dL, *p* < 0.05), total cholesterol (138.90 ± 45.20 vs. 102.51 ± 21.42 mg/dL, *p* < 0.05), and LDL cholesterol (26.54 ± 10.60 vs. 14.17 ± 6.10 mg/dL, *p* < 0.05), and at the same time unchanged HDL cholesterol level compared with the healthy control group. That is clearly the result of disturbed glucose and lipid metabolism accompanying diabetes. Supplementary SGs (both stevioside and rebaudioside A, at low and high doses) were able to normalize the lipid profile, almost to the level comparable with the healthy control rats. Interestingly, the strongest regulatory effect was attributed to supplementary RA at high doses and Metformin. Furthermore, a dose-dependent effect of supplementary SGs on the serum TG was observed, which means that a high dose of SGs (both compounds) significantly reduced this parameter (St: 216.78 ± 80.5, RA: 233.23 ± 30.8 mg/dL, *p* < 0.05), in comparison with the low SGs dose (St: 341.49 ± 165.2 ± 80.5, RA: 274.76 ± 121.5 mg/dL, *p* < 0.05). 

### 3.3. Effects of Supplementary SGs on Other Biochemical Indices

The effects of supplementary SGs on selected blood serum biochemical indices in diabetic rats are presented in Figure 3, Figure 4 and Table 2. The diabetic untreated rats (Db), as expected, had significantly elevated blood glycemia (428.89 ± 131.4 vs. 129.80 ± 7.6 mg/dL, *p* < 0.05), but only slightly (insignificantly) decreased insulinemia (315.45 ± 62.18 vs. 328.44 ± 73.71 mg/dL, *p* < 0.05), markedly increased insulin resistance HOMA-IR index (54.48 ± 24.61 vs. 17.44 ± 3.51), and at the same time decreased HOMA-β (68.68 ± 55.83 vs. 301.58 ± 88.02) and QUICKI indices (0.23 ± 0.009 vs. 0.26 ± 0.006), compared with the healthy control rats. The oral treatment with Metformin (150 mg/kg b.m.) did not make any significant change in these parameters, which shows that the dose was too small to bring about significant improvements in blood glucose and insulin-related indices.

Supplementary SGs (stevioside and rebaudioside A, both at low and high doses) did not significantly affect blood glycemia, though some small (insignificant) decrease was noticed in the group supplemented with a high dose of stevioside (Db + S2). That shows that both SGs (in the applied doses) were unable to significantly improve glucose level and insulin sensitivity indices (HOMA-IR, HOMA-β and QUICKI) in the diabetic rats. Of note, also treatment with Metformin (150 mg/kg b.w.) was ineffective in this regard.

The remaining biochemical blood indices are presented in Table 2. They encompass liver (ALT, AST) and kidney function (UREA, KREA, TP) indices, antioxidant indices (OxLDL, URIC, NO, GPX), as well as appetite regulatory biomarkers (Leptin, Ghrelin and Adiponectin).

Concerning liver function indices, the diabetic rats had significantly elevated serum ALT values (67.00 ± 23.62 vs. 38.15 ± 5.37 U/L, *p* < 0.05), but not AST values, thus AST/ALT ratio decreased (2.22 ± 0.82 vs. 4.59 ± 0.65), compared with the healthy control rats. Supplementary stevioside and rebaudioside A at a high dose (2500 mg/kg b.w.) significantly normalized blood ALT values almost to the level of the healthy control rats, which clearly demonstrates the protective role of these compounds on the liver function.

Kidney function indices included the serum UREA, KREA and TP concentration. It was shown that the diabetic rats (Db) had a significantly increased serum UREA value (47.37 ± 12.22 vs. 31.73 ± 3.46 mg/dL, *p <* 0.05) and at the same time decreased total protein (TP) concentration (5.61 ± 0.67 vs. 6.55 ± 0.09 mg/dL, *p <* 0.05), compared with the healthy control rats. Supplementary stevioside at a high dose (2500 mg/kg b.w.) normalized the serum UREA concentration, while increased TP levels almost to the values of the control healthy rats.

In diabetes, the antioxidant status is usually disturbed by hyperglycemia and metabolic disturbances, thus oxidative stress further exacerbates the systemic damage of tissues and organs. In our experiment, the changes in antioxidant biomarkers, such as the elevated OxLDL (0.69 ± 0.13 vs. 0.56 ± 0.10 µg/mL, *p <* 0.05) and the antioxidant enzyme GPX (178.3 ± 51.3 vs. 121.0 ± 109.7 nmol/min/mL, *p <* 0.05) were found in the diabetic rats, compared with the healthy control rats, which clearly indicates the increased oxidative stress caused by chronic hyperglycemia. 

Supplementary SGs did not affect the antioxidative indices in the diabetic rats, while Metformin was able to decrease these biomarkers in serum of the diabetic rats nearly to the values of the healthy control rats.

The appetite regulatory biomarkers, such as serum ghrelin, leptin and adiponectin, were in general only slightly changed in the diabetic rats, which is presented in Figure 4. Of notice, a relatively wide intragroup distribution of values, both in healthy and diabetic groups was observed. Supplementary rebaudioside A at both doses significantly increased Leptin value (9.42 ± 10.35/13.74 ± 13.92 vs. 0.70 ± 0.14 ng/mL, *p <* 0.05), compared with the untreated diabetic rats. Interestingly enough, Adiponectin/Leptin ratio was markedly decreased in all the diabetic rats supplemented with SGs (at both doses).

### 3.4. Effects of Supplementary SGs on Blood Hematological Indices

The effects of supplementary SGs on blood hematological indices in diabetic rats are presented in Table 3. Experimental factors, such as diabetes and treatments had some variable effects on blood hematology indices in rats. Particularly, diabetes itself did not cause any significant changes in the levels of most of the hematological parameters, except for RDV-CV, WBC, NEUT, LYM, MONO%, MONO values. Diabetic rats treated with Metformin had significantly elevated levels of HGB, EOS%, as compared with the diabetic untreated rats (Db).

Diabetic rats supplemented with stevioside at a low dose (500 mg/kg b.w.) had markedly lower WBC and LYM values, while the group supplemented with a high dose of this compound (2500 mg/kg b.w.) had higher blood HGB and MONO levels, as compared with the Db group.

Diabetic rats supplemented with RA at a higher dose (2500 mg/kg b.w.) had significantly elevated blood HGB and HCT levels, as compared with the Db group. Furthermore, all these changes were contained within the reference values for healthy rats, thus do not indicate any pathological states in the hematopoietic system.

### 3.5. Histopathological Analysis

The histological images of the samples of pancreas, liver, and kidneys are shown in Figure 5, and their description in Figure 6. As presented in Figure 5, the pictures of the tissue samples taken from the healthy control rats did not show any pathological changes, and thus serve as a reference for the diabetic rats. Concerning the pictures of tissues of the diabetic untreated rats (Db), as expected, they show the most severe changes in the pancreas, liver, and kidneys.

Symptoms of degeneration, interstitial lymphocytic pancreatitis and progressive necrosis processes are visible in the pancreatic samples of the untreated diabetic rats (Figure 5A, Db). The architecture and arrangement of cells in the pancreatic islets are clearly disturbed. Pycnotic nuclei and homogeneous masses of cellular cytoplasm are visible, expanded halo-follicular cells around the islets with minor cytoplasmic vacuoles were found. Pycnotic nuclei of the epithelium are visible in pancreatic parenchyma infiltrated by lymphocytes.

The histopathological analysis of the untreated diabetic rats’ liver samples indicates an early stage of acute hepatic steatosis with necrosis, lymphocytic inflammation, and severe fatty degeneration, slight infiltration of mononuclear cells around central veins is visible, connective tissue hyperplasia is observed, while the signs of cell nucleus pycnosis indicate hepatocyte necrosis (Figure 5B, Db).

The kidney samples of the untreated diabetic rats show vacuolization of smooth myocytes of the middle arteries with signs of degeneration, and moderate swelling of renal corpuscle mesangium is visible as well (Figure 5C and 5D, Db).

Concerning the organ samples obtained from the diabetic rats treated with SGs and Met, the above-mentioned pathological alterations are still visible, but to a noticeably lesser degree, specifically the changes are milder to moderate, clearly less severe, as compared with the diabetic untreated rats (Figure 5 and Figure 6).

In particular, in the pancreas, the changes regarding the interstitium, swelling of pancreatic follicle cells, follicular cell necrosis, and the disorganization of the islets architecture are decreased in all the treated groups (Figure 6A, Db + Met, Db + S1, Db + S2, Db + R1 and Db + R2). The congestion and widening of the parenchyma capillaries do not occur in Db + S1 and Db + R1 groups. Groups treated with metformin (Figure 6A, Db + Met), stevioside (Figure 6A, Db + S1, Db + S2), and a lower dose of RA (Figure 6A, Db + R1), did not develop pancreatic alveolar vacuolization. Signs of pancreatic islet cell necrosis are less pronounced compared with the diabetic untreated rats.

Liver histology shows the signs of some hepatocytes necrosis, but clearly less pronounced in the groups treated with metformin (Figure 6B, Db + Met) and a lower dose of RA (Figure 6B, Db + R1). In particular, fine-droplet steatosis is milder in groups treated with metformin (Figure 6B, Db + Met) and stevioside (Figure 6B, Db + S1 and Db + S2). In addition, the groups treated with RA (Figure 6B, Db + R1 and Db + R2) and a higher dose of stevioside (Figure 6B, Db + S2), do not show periportal connective tissue hyperplasia and biliary hyperplasia.

Kidney histology shows some milder changes concerning glomerular deformity and glomerular capillary atrophy in the groups treated with stevioside (Figure 6C, Db + S1 and Db + S2). The groups receiving higher doses of stevioside (Figure 6C, Db + S2) and rebaudioside A (Figure 6C, Db + R2), did not develop the thickening of the basal membranes of the glomeruli. Central arteriolar hypertrophy and perivascular edema are less pronounced in the groups treated with metformin (Figure 6C, Db + Met), a lower dose of stevioside (Figure 6C, Db + S1), and a higher dose rebaudioside A (Figure 6C, Db + R2). The renal tubular degeneration is clearly decreased in all the treated groups.

## 4. Discussion

The present study investigated the effects of supplementary SGs (stevioside and rebaudioside A) on diabetic rats with particular attention to lipid metabolism, to evaluate its regulatory or therapeutic potential in chronic hyperglycemia.

The overall nutritional indices showed that rats from the diabetic groups had a significantly higher feed intake despite eating a high-calorie diet, but lower body mass gain and feeding efficacy ratio (FER) compared to the healthy control group. This was caused by the metabolic disturbances associated with hyperglycemia induced by high fat feeding and STZ injection. Supplementary SGs markedly decreased average daily feed intake in a dose-dependent fashion in the diabetic rats, nearly to the level of the healthy rats. This shows that SGs affect appetite control (vide infra). Similar effects were reported by other authors [41,42].

The biological control of appetite is a very complex phenomenon that involves a variety of factors, including a host of nutrient sensors, endocrine factors, and neural signals, while the key regulatory hormones involved in appetite control are leptin, ghrelin, and adiponectin [43].

In our experiment, the diabetic rats fed HF diet supplemented with SGs had an increased blood leptin level (insignificantly for stevioside, but significantly for rebaudioside A, at both doses) that corresponded with a decreased feed intake compared with the untreated diabetic group. The circulating adiponectin levels were only slightly decreased in the diabetic rats, but the adiponectin/leptin ratio was markedly decreased in all the diabetic rats supplemented with SGs, while insulin resistance indices were not affected by the treatment. This is in agreement with previous observations [44].

High-fat diet (HFD) consumption increases hepatic glucose output and decreases glucose uptake in peripheral tissues, resulting in hyperglycemia. This increases insulin secretion and consequently activates hepatic lipogenesis and cholesterol excretion, leading to disturbances in lipid metabolism (evidenced by hyperlipidemia) [45]. These disturbances commonly include elevated levels of triacylglycerol (TG) and low-density lipoprotein (LDL cholesterol), as well as decreased levels of high-density lipoprotein (HDL cholesterol) in blood. Prolonged dyslipidemia leads to serious cardiovascular system complications, thus regulation of lipid profile is crucial in the treatment of diabetes.

In diabetes, the concentration of serum lipids is usually elevated, which represents a high-risk factor for coronary heart disease. Under reasonable conditions, insulin activates the enzyme lipoprotein lipase, which hydrolyzes triglycerides. However, in a diabetic state, lipoprotein lipase is not activated sufficiently due to insulin deficiency resulting in hypertriglyceridemia [46].

In our experiment, the diabetic untreated rats (Db) had significantly elevated blood lipids, except HDL cholesterol, which is clearly the result of disturbed glucose metabolism accompanying diabetes. Treatment with SGs (both stevioside and rebaudioside A, at low and high doses) normalized the lipid profile, almost to the levels comparable with the healthy control rats. Analysis of variance showed that the stronger blood TG lowering effect (by 27%) was attributed to a diet supplemented with high SGs dose (2500 mg/kg b.m.), independently on the type of steviol glycoside, compared with the low SGs dose diet.

The mechanisms of the lipid-lowering effect of steviol glycosides have not been fully understood and appear to be manifold. Holvoet et al. (2015) found that stevioside, rebaudioside A and steviol attenuated hepatic steatosis that could be explained by improved glucose metabolism, fat catabolism, bile acid metabolism and lipid storage and transport [47]. They identified PPARs as important regulators and observed differences in effects on insulin resistance, inflammation and oxidative stress between stevia-derived compounds.

Some studies reported that stevia compounds act like a moderate PPARα activator [48,49]. Rotimi et al. (2018) found that the lipid profile improving effect of stevioside is related to the inhibition of G-protein-coupled receptor kinase, which is upregulated in dyslipidemia [50]. Our study revealed that pure SGs poses significant lipid regulating properties in the case of hyperlipidemia accompanying diabetes.

For patients suffering from diabetes, the most critical challenge is the normalization of blood glucose levels to prevent glucotoxicity and related complications. The beneficial effects of stevia and its extracts or isolated compounds on blood glucose levels have been reported in some animal studies [15,50,51,52] as well as in a few human trials [53,54]. However, the antidiabetic effects depended mainly on the form of stevia, its dose and duration of treatment used in a particular study [55].

It was found by Jeppesen et al. (2000) that both stevioside and steviol increased insulin secretion from the cells [56]. Philippaert et al. (2017) reported that steviol glycosides potentiate the TRPM5-mediated glucose-induced depolarizing current in pancreatic beta cells, enhancing insulin secretion and resulting in lower long-term serum glucose levels [57]. It was observed by Prata et al. (2017) that stevioside increases GLUT4 translocation due to the activation of the PI3K/Akt pathway [58]. Stevioside’s inhibitory effect on the activity of phosphoenolpyruvate carboxykinase (PEPCK), a rate-limiting enzyme required for gluconeogenesis, was discovered by Chen et al. (2005) [51].

Concerning the metabolism of steviol glycosides, it is known that upon consumption, steviol glycosides are not digested in the gut of humans and animals, but hydrolyzed to aglycone steviol and glucose moiety in the colon by intestinal microflora (the Bacteroidaceae family) [59]. Glucose is utilized by the colon microbiota, while some free steviol is excreted in feces, and some is absorbed into the bloodstream and metabolized by the liver into steviol glucuronide [60]. This process requires glucose delivery and conversion to glucuronic acid for coupling with steviol. Steviol glucuronide is further excreted in urine [61,62]. Recently, Myint et al. (2020) reported that the antidiabetic effect of SGs is attributed to steviol present in a glycoside [63]. The amount of steviol is higher in stevioside compared with rebaudioside A (39.6% vs. 32.9%, respectively), thus a stronger effect might be expected for stevioside compared with rebaudioside A. In our study, supplementary SGs (stevioside and rebaudioside A) did not affect blood glucose levels in diabetic rats, but some noticeable decrease (statistically insignificant) of blood glucose concentration was observed in the group treated with stevioside at a high dose (2500 mg/b.w.) that seem to correspond with the higher content of steviol in the molecule, as compared with rebaudioside A.

In this study, serum insulin levels were not significantly different in the experimental groups (both diabetic and healthy), probably due to incomplete damage of the pancreas islets, that were still able to secrete a sufficient amount of insulin. However, insulin resistance indices (HOMA-IR, HOMA-β, QUICKI) confirmed an overt systemic insulin resistance in all the diabetic groups. Neither supplementary SGs nor Met were able to improve insulin sensitivity indices in the diabetic rats. This is in agreement with some experimental studies and a recent meta-analysis of randomized controlled trials applying pure stevia glycosides that showed that their antidiabetic potential is low [64,65]. On the other hand, an appreciable antidiabetic potential of stevia was reported when fractions of the stevia plant containing other phytochemicals (i.e., polyphenols, antioxidants, fiber) were applied [53,66,67].

Long-term high-fat diet intake causes ectopic fat deposition in several organs other than the adipose tissue. In particular, hepatic fat accumulation arises from increased circulation fatty acid uptake and decreased β-oxidation in the liver, leading to non-alcoholic fatty liver disease (NAFLD). Aspartate transaminase (AST) and alanine transaminase (ALT) are considered biomarkers of liver health, which is commonly deteriorated in diabetes and feeding HF diet. NAFLD following HF diet is denoted by the elevated leakage of liver enzymes (AST, ALT) from hepatocytes into the blood. In our experiment, the blood ALT level was significantly elevated, AST was unchanged, while the AST/ALT ratio was decreased by approximately 50% in the diabetic untreated rats, which indicates liver cell damage caused by fatty acids uptake and glucotoxicity (chronic hyperglycemia) in rats. Supplementary stevioside and rebaudioside A (at high doses of 2500 mg/kg b.w.), but not metformin, were able to ameliorate ALT levels in the diabetic rats, almost to the levels of the healthy control rats. Also, the AST/ALT values were significantly increased, except in the Db + R2 group. These effects seem to be consistent with the histology images of the liver, where less pathological changes were observed in the livers of diabetic rats supplemented with both SGs. Similar liver-protecting effects of supplementary stevia-derived compounds were reported by other authors [47,67,68,69,70].

In diabetes, the kidneys are challenged to excrete excessive amounts of glucose that leads to gradual deterioration of the organ structure and functions. Abnormal levels of blood/urine indices, such as blood UREA, KREA and TP indicate kidney dysfunction in diabetes.

In our experiment, blood UREA level was significantly elevated, while total protein TP levels were decreased in the diabetic untreated rats. Only supplementary stevioside at a high dose (2500 mg/kg b.w.) normalized these indices almost to the levels of the healthy control rats, which corresponds with a noticeable improvement in the histology of kidney images (vide infra). This clearly indicates renoprotective properties of this compound in diabetes. Mild renoprotective effects of stevia have been previously reported by other authors [71,72,73,74].

In diabetes, the antioxidant status is usually disturbed by chronic hyperglycemia, while concurring oxidative stress further exacerbates the systemic damage of tissues and organs. Overproduction of free radicals and defect of antioxidants’ protection involved in the pathogenesis of diabetes was reported in various experimental trials [75,76]. The ability of stevioside to counteract free radicals and reduce oxidative damage has been reported in previous studies [77,78,79].

In our study, serum OxLDL and glutathione peroxidase GPX levels were elevated, which clearly indicates the increased oxidative stress caused by chronic hyperglycemia. Supplementary Metformin was able to decrease these biomarkers in the diabetic rats nearly to the values of the healthy control, whereas supplementary pure SGs did not affect the antioxidant status in the diabetic rats. This result is in agreement with the observation that stevia extracts may attenuate oxidative stress [80], but the action is not dependent on steviol glycosides, but related to other phytochemicals present in the plant [81].

There are limitations to this study. We used an artificial model of diabetes (streptozotocin-induced), which has been widely used in this kind of research, however there are differences in the mechanisms causing insulin resistance and T2DM in human. Also the tested doses of supplementary steviol glucosides were relatively high and given in a short-term setting. Furthermore, we did not analyze other indices (i.e., GTT, ITT assays, other lipid metabolic indices) due to limited resources. Future study will be performed in our lab that should include those parameters. The above mentioned limitations cause that the data should be interpreted in caution.

## 5. Conclusions

This study showed that supplementation of steviol glycosides (stevioside and rebaudioside A) does not improve blood glucose, insulin, insulin resistance indices, or antioxidant biomarkers, but normalize hyperlipidemia, significantly improve liver and kidney function indices, affect feed intake and some appetite control biomarkers, and attenuate damage in pancreatic, hepatic, and renal tissue in diabetic rats.

These effects allow us to conclude that supplementation of both stevioside and rebaudioside A could provide an alternative therapy for T2DM. Further studies, including clinical trials, are warranted to confirm these effects in humans, and fully understand the mechanisms of action on the molecular level.

## Figures and Tables

**Figure 1 nutrients-13-00112-f001:**
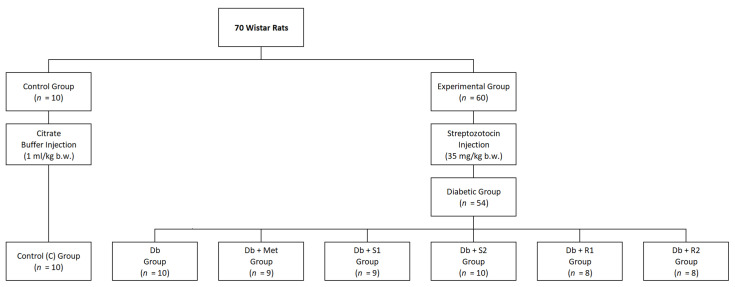
The allocation of rats to individual groups. Control group (C): healthy rats receiving AIN-93M diet, Db group: diabetic rats receiving high-fat (HF) diet, Db + metformin (Met) group: diabetic rats receiving HF diet with 0.15% metformin, Db + S1 group: diabetic rats receiving HF diet with 0.5% stevioside, Db + S2 group: diabetic rats receiving HF diet with 2.5% stevioside, Db + R1 group: diabetic rats receiving HF diet with 0.5% rebaudioside A, Db + R2 group: diabetic rats receiving HF diet with 2.5% rebaudioside A.

**Figure 2 nutrients-13-00112-f002:**
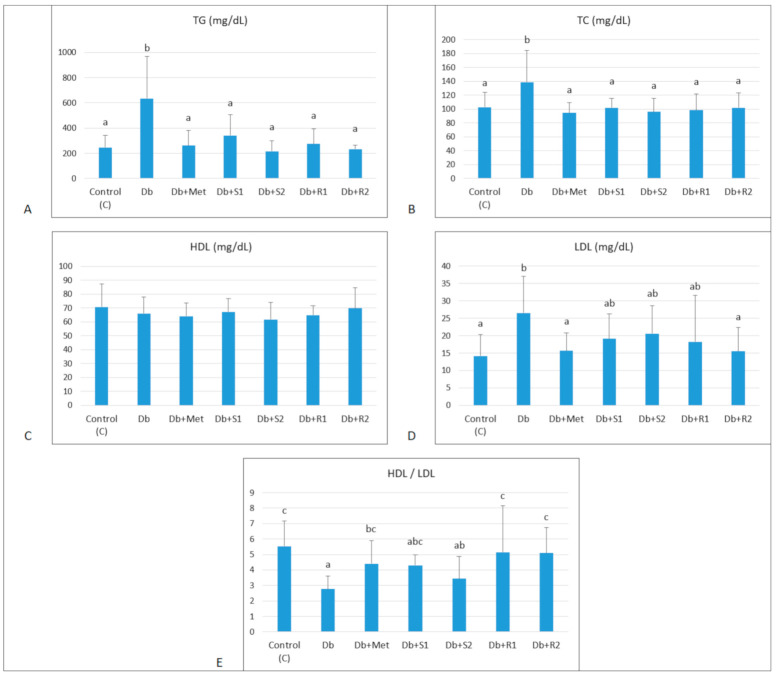
Effect of diet and supplementary SGs on lipid profile indices of rats. (**A**) TG: serum triacylglycerols, (**B**) TC: serum total cholesterol, (**C**) HDL: serum high-density lipoproteins, (**D**) LDL: serum low density lipoproteins and (**E**) HDL/LDL ratio in serum of experimental rats. Data are presented as mean ± standard deviation (M ± SD). Values with different letters (a–d) show statistically significant differences (*p* < 0.05, a < b, Fisher’s LSD test).

**Figure 3 nutrients-13-00112-f003:**
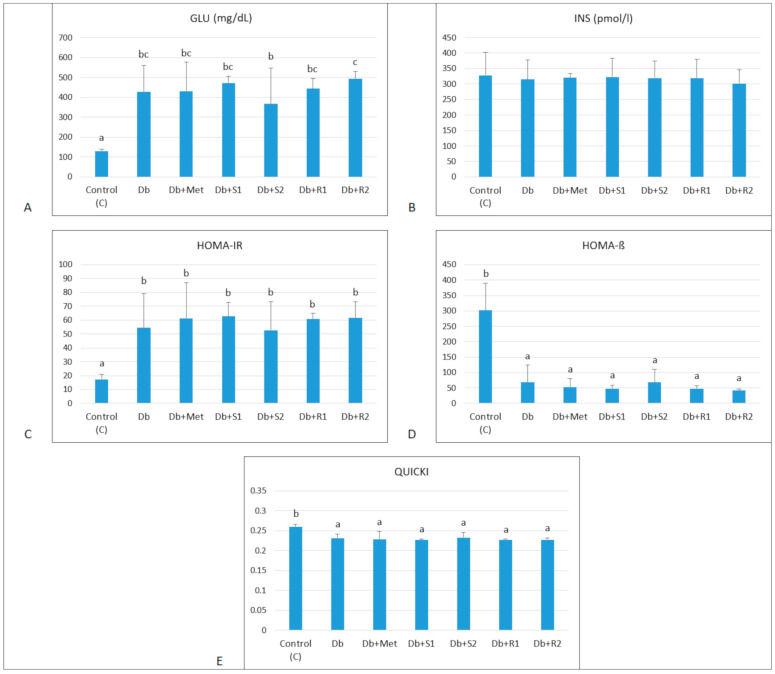
Effect of diet and supplementary SGs on (**A**) GLU: fasting glucose concentration, (**B**) INS: insulin concentration, (**C**) HOMA-IR: insulin resistance index, (**D**) HOMA-β: β-cell function index and (**E**) QUICKI: the quantitative insulin sensitivity check index in rats. Data are presented as mean ± standard deviation (M ± SD). Values with different letters (a–c) show statistically significant differences (*p* < 0.05, a < b, Fisher’s LSD test).

**Figure 4 nutrients-13-00112-f004:**
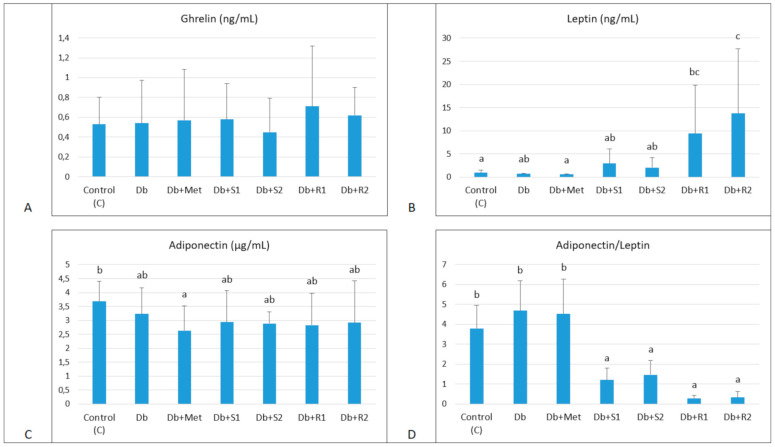
Effect of diet and supplementary SGs on appetite regulatory biomarkers. (A) Ghrelin: total ghrelin, (B) Leptin, (C) Adiponectin, (D) Adiponectin/Leptin: adiponectin to leptin ratio in serum of experimental rats. Data are presented as mean ± standard deviation (M ± SD). Values with different letters (a–c) show statistically significant differences (*p <* 0.05, a < b, Fisher’s LSD test).

**Figure 5 nutrients-13-00112-f005:**
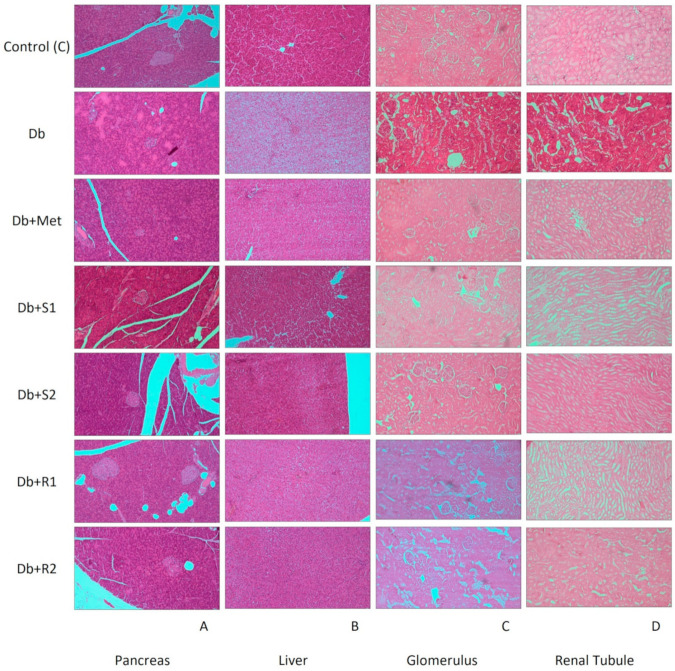
Effect of diet and supplementary SGs on histological alterations of pancreas, liver, and kidneys. Internal cross-sections of (**A**) pancreas, (**B**) liver, (**C**) kidneys—glomerulus, and (**D**) kidneys—renal tubules (10× magnified).

**Figure 6 nutrients-13-00112-f006:**
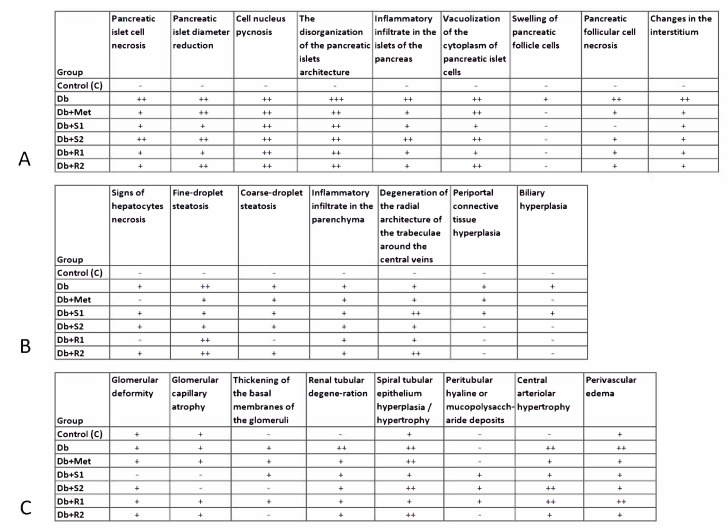
Effect of diet and supplementary SGs on histological alterations in organs of rats: (**A**) pancreas, (**B**) liver, and (**C**) kidneys. Symbols indicate the severity of the changes in each group. Visible pathological changes: no (-), mild (+), moderate (++), severe (+++).

**Table 1 nutrients-13-00112-t001:** Effect of diet and supplementary steviol glycosides (SGs) on overall growth indices and relative organ mass in rats.

Parameter	Control (C)	Experimental Groups					
		Db	Db + Met	Db + S1	Db + S2	Db + R1	Db + R2
Avg FI (g/day/rat)	27.20 ± 1.61 abc	29.94 ± 3.94 de	31.62 ± 5.01 e	28.57 ± 3.86 cd	26.26 ± 4.30 ab	27.26 ± 4.16 bc	25.54 ± 3.23 a
Initial b.w. (g)	469.8 ± 43.77 a	502.8 ± 36.27 b	502.8 ± 37.44 ab	503.5 ± 31.32 b	510.2 ± 30.21 b	503.0 ± 36.82 ab	497.4 ± 27.80 b
Final b.w. (g)	543.4 ± 51.50 b	473.4 ± 46.86 a	488.7 ± 51.92 a	482.8 ± 64.13 a	487.9 ± 49.80 a	450.9 ± 58.79 a	476.5 ± 27.26 a
FER	1.71 ± 0.46 b	0.02 ± 1.67 a	0.33 ± 0.31 a	0.04 ± 1.49 a	−0.14 ± 0.93 a	−0.25 ± 0.32 a	−0.04 ± 0.83 a
Liver (% b.m.)	3.19 ± 0.31 a	3.50 ± 0.26 ab	3.85 ± 0.43 c	3.54 ± 0.37 bc	3.49 ± 0.35 ab	3.57 ± 0.40 bc	3.68 ± 0.33 bc
Lungs (% b.m.)	0.32 ± 0.04 a	0.41 ± 0.07 b	0.41 ± 0.07 b	0.42 ± 0.06 b	0.40 ± 0.06 b	0.45 ± 0.08 b	0.40 ± 0.04 b
Kidneys (% b.m.)	0.63 ± 0.07 a	0.95 ± 0.17 b	0.94 ± 0.12 b	0.94 ± 0.20 b	0.85 ± 0.16 b	0.98 ± 0.19 b	0.94 ± 0.04 b
Testes (% b.m.)	0.76 ± 0.10 a	0.81 ± 0.19 ab	0.82 ± 0.15 ab	0.85 ± 0.07 ab	0.84 ± 0.14 ab	0.90 ± 0.17 b	0.91 ± 0.11 b
Spleen (% b.m.)	0.16 ± 0.01 a	0.18 ± 0.03 ab	0.19 ± 0.03 b	0.18 ± 0.05 ab	0.19 ± 0.03 b	0.19 ± 0.02 ab	0.19 ± 0.03 b
Heart (% b.m.)	0.26 ± 0.02 a	0.31 ± 0.03 b	0.30 ± 0.01 b	0.30 ± 0.03 b	0.30 ± 0.02 b	0.31 ± 0.03 b	0.31 ± 0.03 b
Brain (% b.m.)	0.39 ± 0.05 a	0.43 ± 0.05 ab	0.44 ± 0.05 b	0.42 ± 0.04 ab	0.45 ± 0.04 b	0.47 ± 0.08 b	0.43 ± 0.02 ab

Note. Data are presented as mean ± standard deviation (M ± SD). Mean values with different letters (a–d) in rows show statistically significant differences (*p* < 0.05, a < b, Fisher’s LSD test). Avg FI: average feed intake after inducing hyperglycemia, Initial b.w.: initial body weight after inducing hyperglycemia, Final b.w.: final body weight, FER: feed efficiency ratio after inducing hyperglycemia.

**Table 2 nutrients-13-00112-t002:** Effect of diet and supplementary SGs on selected blood biochemical indices in rats.

Parameter	Control (C)	Experimental Groups					
		Db	Db + Met	Db + S1	Db + S2	Db + R1	Db + R2
ALT (U/l)	38.15 ± 5.37 a	67.00 ± 23.62 c	67.14 ± 21.12 c	59.30 ± 18.64 bc	48.46 ± 11.60 ab	57.94 ± 15.75 bc	48.20 ± 13.25 ab
AST (U/l)	174.7 ± 33.0 abc	133.4 ± 19.8 a	181.6 ± 41.9 bc	172.2 ± 43.5 abc	181.0 ± 65.7 c	188.6 ± 73.7 c	133.7 ± 25.7 ab
AST / ALT	4.59 ± 0.65 c	2.22 ± 0.82 a	2.61 ± 0.45 ab	3.04 ± 0.78 b	3.44 ± 0.95 b	3.26 ± 1.06 b	2.98 ± 1.11 ab
UREA (mg/dL)	31.73 ± 3.46 a	47.37 ± 12.22 c	39.68 ± 7.70 abc	45.31 ± 6.31 bc	39.13 ± 11.84 ab	45.30 ± 9.35 bc	40.08 ± 8.06 abc
KREA (mg/dL)	0.30 ± 0.00	0.30 ± 0.00	0.30 ± 0.00	0.28 ± 0.07	0.30 ± 0.00	0.30 ± 0.00	0.28 ± 0.07
TP (g/dl)	6.55 ± 0.09 c	5.61 ± 0.67 ab	5.67 ± 0.31 ab	5.68 ± 0.50 ab	6.06 ± 0.35 bc	5.43 ± 0.38 a	5.84 ± 0.73 ab
OxLDL (µg/mL)	0.56 ± 0.10 a	0.69 ± 0.13 bc	0.59 ± 0.05 ab	0.67 ± 0.16 bc	0.67 ± 0.09 bc	0.73 ± 0.17 c	0.70 ± 0.10 bc
URIC (mg/dL)	1.59 ± 0.28 ab	1.45 ± 0.28 ab	1.71 ± 0.35 b	1.47 ± 0.59 ab	1.52 ± 0.50 ab	1.59 ± 0.13 ab	1.32 ± 0.34 a
NO (µmol/L)	0.97 ± 0.82 a	3.27 ± 3.08 ab	3.76 ± 2.99 ab	2.48 ± 1.69 ab	2.97 ± 1.14 ab	4.00 ± 3.35 b	5.06 ± 5.37 b
GPX (nmol/min/mL)	121.0 ± 109.7 a	178.3 ± 51.3 b	154.2 ± 61.5 ab	203.0 ± 24.6 b	197.0 ± 47.1 b	203.2 ± 32.0 b	173.6 ± 64.9 ab

Note. Data are presented as mean ± standard deviation (M ± SD). Mean values with different letters (a–c) in rows show statistically significant differences (*p* < 0.05, a < b, Fisher’s LSD test). ALT: alanine transaminase, AST: aspartate transaminase, AST / ALT: alanine transaminase to aspartate transaminase ratio, UREA: urea/carbamide, KREA: creatinine, TP: total protein, OxLDL: oxidized low density lipoprotein, URIC: uric acid, NO: nitric oxide, GPX: glutathione peroxidase.

**Table 3 nutrients-13-00112-t003:** Effect of diet and supplementary SGs on blood Hematological indices in rats.

Parameter	Control (C)	Experimental Groups					
		Db	Db + Met	Db + S1	Db + S2	Db + R1	Db + R2
RBC (T/l)	9.10 ± 0.46	8.89 ± 0.41	8.92 ± 0.46	9.08 ± 0.62	9.12 ± 0.38	8.84 ± 0.49	9.10 ± 0.45
HGB (g/dl)	15.83 ± 0.38 a	15.83 ± 0.69 a	16.41 ± 0.35 b	16.25 ± 0.46 ab	16.34 ± 0.45 b	16.23 ± 0.46 ab	16.55 ± 0.77 b
HCT (%)	46.70 ± 2.43 a	46.30 ± 2.21 a	47.89 ± 1.01 ab	48.34 ± 3.35 ab	47.83 ± 1.71 ab	47.38 ± 2.44 ab	49.03 ± 2.26 b
MCV (fl)	51.34 ± 1.42 a	52.15 ± 2.16 ab	53.84 ± 2.28 b	53.21 ± 2.10 b	52.52 ± 2.03 ab	53.56 ± 2.56 b	53.95 ± 1.40 b
MCH (pg)	17.60 ± 0.57 a	17.82 ± 0.70 ab	18.44 ± 0.63 bc	18.53 ± 0.21 c	17.96 ± 0.55 abc	18.35 ± 0.97 bc	18.25 ± 0.82 bc
MCHC (g/dL)	34.30 ± 0.60	34.18 ± 0.47	34.31 ± 0.78	34.27 ± 0.87	34.17 ± 0.90	33.91 ± 0.35	33.78 ± 0.82
PLT (G/l)	914.60 ± 106.2	843.30 ± 122.4	845.67 ± 114.5	961.44 ± 123.6	867.00 ± 171.5	855.00 ± 162.3	917.63 ± 166.6
RDW-CV (%)	14.16 ± 0.59 a	15.16 ± 1.07 bc	14.96 ± 0.94 abc	15.59 ± 1.04 c	15.15 ± 1.21 bc	14.98 ± 0.90 abc	14.43 ± 0.36 ab
WBC (G/l)	8.55 ± 2.20 a	10.70 ± 1.61 b	8.79 ± 2.58 ab	8.44 ± 1.81 a	9.39 ± 1.96 ab	9.10 ± 2.51 ab	8.75 ± 1.69 ab
NEUT% (%)	25.42 ± 7.37 a	27.82 ± 7.32 ab	34.83 ± 8.38 c	31.31 ± 5.62 abc	28.84 ± 4.27 abc	34.55 ± 6.37 bc	33.29 ± 11.20 bc
NEUT (G/l)	1.58 ± 0.31 a	2.71 ± 0.61 b	2.97 ± 0.85 b	2.61 ± 0.59 b	2.70 ± 0.66 b	3.12 ± 0.98 b	2.91 ± 1.03 b
LYM% (%)	64.24 ± 7.70 b	59.99 ± 9.30 ab	53.07 ± 10.14 a	54.39 ± 8.22 a	59.95 ± 6.14 ab	52.74 ± 7.40 a	53.68 ± 14.27 a
LYM (G/l)	5.43 ± 1.25 ab	6.19 ± 1.91 b	4.78 ± 2.10 ab	4.61 ± 1.34 a	5.65 ± 1.33 ab	4.83 ± 1.69 ab	4.70 ± 1.57 ab
MONO% (%)	6.62 ± 1.26 a	8.70 ± 2.74 b	7.29 ± 1.69 ab	8.91 ± 2.45 b	7.02 ± 2.06 ab	8.48 ± 2.20 ab	7.34 ± 2.49 ab
MONO (G/l)	0.52 ± 0.14 a	0.88 ± 0.37 b	0.62 ± 0.17 a	0.75 ± 0.26 ab	0.65 ± 0.22 a	0.67 ± 0.16 ab	0.65 ± 0.29 ab
EOS% (%)	3.12 ± 0.69 ab	2.78 ± 0.93 a	3.88 ± 1.27 b	3.25 ± 1.46 ab	2.93 ± 0.36 ab	3.36 ± 1.56 ab	2.54 ± 0.27 a
EOS (G/l)	0.26 ± 0.06 a	0.28 ± 0.12 ab	0.35 ± 0.17 ab	0.26 ± 0.10 ab	0.31 ± 0.13 ab	0.30 ± 0.15 ab	0.41 ± 0.28 b
BAS% (%)	0.60 ± 0.23 a	0.71 ± 0.34 ab	0.93 ± 0.33 b	0.90 ± 0.36 b	0.88 ± 0.32 ab	0.88 ± 0.35 ab	0.90 ± 0.32 ab

Note. Data are presented as mean ± standard deviation (M ± SD). Mean values with different letters (a–c) in rows show statistically significant differences (*p <* 0.05, a < b, Fisher’s LSD test). RBC: red blood cell, HGB: blood hemoglobin, HCT: hematocrit, MCV: mean corpuscular volume, MCH: mean corpuscular hemoglobin, MCHC: mean corpuscular hemoglobin concentration, PLT: platelets, RDW-CV: red blood cell distribution width, WBC: white blood cells, NEUT: neutrophils, LYM: lymphocytes, MONO: monocytes, EOS: eosinophils, BAS: basophils.

## Data Availability

The data presented in this study are available on request from the corresponding author.

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
