# Peer review of "Steviol Glycosides Supplementation Affects Lipid Metabolism in High-Fat Fed STZ-Induced Diabetic Rats"

_nutrients, 2020, doi:10.3390/nu13010112_

Round 1

Reviewer 1 Report

The manuscript is very interesting and well organized. The methodology is original and findings are relevant. In particular, the study adds important elements to clarify the potential protective effect of supplementary steviol glycosides on high-fat fed streptozotocin-induced diabetic rats. Nevetheless, some points need to be addressed.

  1. Fasting blood glucose and fasting insulin as well as and lipid profile were determined and serum inflammatory factors were analyzed. Did you perform glucose tolerance test (GTT) or insulin tolerance test (ITT) to measure the glycometabolism of the diabetic rats at the end of the treatment? What about circulating free fatty acids?
  2. Given your interesting results in this model, the expression of PPARγ protein and PI3K/Akt signaling pathway related proteins in adipose tissue  from tese rats would add some significant knowledge about the mechanism by which steviol glycosides exerts its effects. Did you analyze any marker in adipose tissue or skeletal muscle?
  3. Tables are difficult to read, it would be useful to add graph for the most relevant results. 

Author Response

Response to Reviewer 1 Comments

• Point 1 & 2:

Fasting blood glucose and fasting insulin as well as and lipid profile were determined and serum inflammatory factors were analyzed. Did you perform glucose tolerance test (GTT) or insulin tolerance test (ITT) to measure the glycometabolism of the diabetic rats at the end of the treatment? What about circulating free fatty acids?

Given your interesting results in this model, the expression of PPARγ protein and PI3K/Akt signaling pathway related proteins in adipose tissue from tese rats would add some significant knowledge about the mechanism by which steviol glycosides exerts its effects. Did you analyze any marker in adipose tissue or skeletal muscle?

• Response 1 & 2:
We do appreciate your valuable comments and suggestions. In this experiment we focused on a variety of overall and specific indices, including the blood glucose, insulin and insulin related indices, lipid profile and other biomarkers. We agree that in light of the obtained results, it would be interesting to perform GTT, ITT assays and circulating free fatty acids levels, as well as the expression of PPARγ protein and PI3K/Akt signaling pathway related proteins in adipose tissue, however in this experiment, due to limited time and resources, we focused on the most relevant in the first approach. In the following experiment of this kind, we will extend the scope of analyses to include the above mentioned assays, to get some further knowledge about the mechanisms of steviol glycosides action.

• Point 3:
Tables are difficult to read, it would be useful to add graph for the most relevant results.

• Response 3:
Following your advice, we presented the appetite regulatory markers (GHR, LEP, ADIPOQ and ADIPOQ/LEP) in a separate figure, excluding them from the table 3, thus the table looks more transparent.

Reviewer 2 Report

Comments:

a. Could the authors please clarify the ethics for the animal experiments?

b. How were the numbers of animals determined for each experiment i.e. how was the study powered?

c. Could the authors justify the use of parametric statistics?

Author Response

Response to Reviewer 2 Comments

We do appreciate your valuable comments and suggestions.
All Your remarks were considered and duly implemented.

• Point a:

Could the authors please clarify the ethics for the animal experiments?

• Response a:
Certainly, the protocol for experimentation on animals used in this study was reviewed, evaluated and approved by the Local Ethical Commission (Approval No. 31/2019). The information about the ethics for this experiment is included in 2.3. Experimental Protocol subsection.

• Point b:
How were the numbers of animals determined for each experiment i.e. how was the study powered?

• Response b:
The number of animals per experimental group was established based on a standard approach calculation (an expected blood glucose level) as follows: for a two-sided t-test with alpha=0.05 and the standard deviation of 0.25 (25%) for the power of 0.80, the sample size is 9. However we used 10 animals per experimental group, taking into account a possible loss of diseased animals (10% attrition). (Inference for Means: Comparing Two Independent Samples; http://www.stat.ubc.ca /~rollin/stats/ssize/n2.html)

• Point c:
Could the authors justify the use of parametric statistics?

• Response c:
The parametric statistics was applied prior checking for all the data normality of distribution (using the Shapiro-Wilk test). Factually almost all the parameters were normally distributed, except for 2 biomarkers, that were duly log-transformed to give a normal distribution. Therefore the parametric statistics was applied.

Reviewer 3 Report

This paper assesses the effect of steviol glycosides on diabetic Wistar rat 

Abstract

Line 25 to 26- "diminished tissular damage" Please use another appropriate term

Line 26 to 27- rephrase the sentence

Introduction

Line 33 to 34- rephrase, confusing

Line 40 to 41- macrovascular and microvascular complications - separate them and explain

Line 42 to 44 - confusing, rephrase

Line 47 to 49-The sentence does not make sense, rephrase

Method

Line 201- What is the difference between ANOVA1 and ANOVA2

Results

Table 1, 2, 3 and Figures 2 and 3- what is "a", "b" and "c"

Line 331 to 332- What do the author mean by "scatter of values"

Figure 5 is very busy and confusing, maybe summarize or calculate overall score

Discussion

Maybe integrate discussion from paragraph line 398 to 406 with paragraph line 630 to 654

Author Response

Response to Reviewer 3 Comments

We do appreciate your valuable comments and suggestions.
All Your remarks were considered and duly implemented.

• Point 1:

Abstract
Line 25 to 26- "diminished tissular damage" Please use another appropriate term.
Line 26 to 27- rephrase the sentence

• Response 1:
The phrase: “reduced tissular damage” was used instead.
The sentence was rephrased.

• Point 2:
Introduction
Line 33 to 34- rephrase, confusing
Line 40 to 41- macrovascular and microvascular complications - separate them and explain
Line 42 to 44 - confusing, rephrase
Line 47 to 49-The sentence does not make sense, rephrase

• Response 2:
Line 33 to 34 was rephrased.
Line 40 to 41 was separated and shortly exemplified.
Line 42 to 44 was rephrased.
Line 47 to 49 was rephrased.

• Point 3:
Method
Line 201- What is the difference between ANOVA1 and ANOVA2

• Response 3:
The sentence was corrected into: “The analysis of variance (ANOVA) and Fisher’s LSD post hoc test (p<0.05) were used to evaluate the level of statistical significance of differences in individual parameters between the Groups”.

• Point 4:
Results
Table 1, 2, 3 and Figures 2 and 3- what is "a", "b" and "c"
Line 331 to 332- What do the author mean by "scatter of values"
Figure 5 is very busy and confusing, maybe summarize or calculate overall score.

• Response 4:
It was commented under the section: 2.7 Statistical Analyses
The data are presented in tables and figures. The statistically significant differences in values between experimental groups were marked with letters: a, b, c etc. Mean values with unlike letters in rows show statistically significant differences (p < 0.05, Fisher’s LSD test).

Line 331 to 332 was changed into: “Of notice, a relatively wide intragroup distribution of values, both in healthy and Diabetic Groups was observed”.

Figure 5 was shortened and reformatted.

• Point 5:
Discussion
Maybe integrate discussion from paragraph line 398 to 406 with paragraph line 630 to 654

• Response 4:
It was integrated as suggested.

Round 2

Reviewer 2 Report

The authors have satisfactorily responded to my comments.

Author Response

We would like to thank you kindly.
We do appreciate your valuable comments and suggestions.